# Probing the Interaction between Isoflucypram Fungicides and Human Serum Albumin: Multiple Spectroscopic and Molecular Modeling Investigations

**DOI:** 10.3390/ijms241512521

**Published:** 2023-08-07

**Authors:** Xiangshuai Li, Xiaojing Yan, Daibin Yang, Shuning Chen, Huizhu Yuan

**Affiliations:** State Key Laboratory for Biology of Plant Diseases and Insect Pests, Institute of Plant Protection, Chinese Academy of Agricultural Sciences, Beijing 100193, China; lixiangshuaijiayou@163.com (X.L.); chenshuning@caas.cn (S.C.)

**Keywords:** isoflucypram, mechanism, human serum albumin, multiple spectroscopic methods, docking simulation

## Abstract

To better understand the potential toxicity risks of isoflucypram in humans, The interaction between isoflucypram and HSA (human serum albumin) was studied through molecular docking, molecular dynamics simulations, ultraviolet–visible absorption, fluorescence, synchronous fluorescence, three-dimensional fluorescence, Fourier transform infrared spectroscopies, and circular dichroism spectroscopies. The interaction details were studied using the molecular docking method and molecular dynamics simulation method. The results revealed that the effect of isoflucypram on human serum albumin was mixed (static and dynamic) quenching. Additionally, we were able to obtain important information on the number of binding sites, binding constants, and binding distance. The interaction between isoflucypram and human serum albumin occurred mainly through hydrogen bonds and van der Waals forces. Spectroscopic results showed that isoflucypram caused conformational changes in HSA (human serum albumin), in which the α-helix was transformed into a β-turn, β-sheet, and random coil, causing the HSA structure to loosen. By providing new insights into the mechanism of binding between isoflucypram and human serum albumin, our study has important implications for assessing the potential toxicity risks associated with isoflucypram exposure.

## 1. Introduction

Pesticides are widely used in agriculture to protect crops from pests and diseases [1]. Their use has led to an increasing concern about their potential adverse effects on human health and the environment [2]. Toxicity assessments for humans are essential for any pesticide [3,4]. Isoflucypram (Figure 1), the first representative of a newly installed subclass of SDHIs (succinate dehydrogenase inhibitors) within the FRAC (Fungicide Resistance Action Committee) family, has shown remarkable and long-lasting efficacy against major foliar diseases in cereals, as reported by Desbordes et al. [5]. However, the essential role of the SDH (succinate dehydrogenase) complex as the molecular target of SDHIs in the energy metabolism of almost all extant eukaryotes, along with the lack of species specificity of these fungicides, raise concerns about their toxicity toward off-target organisms and, more generally, toward the environment. Yanicostas et al. [6] reviewed the current knowledge regarding the toxicity of nine commonly used SDHI fungicides, including isoflucypram, towards zebrafish (*Danio rerio*). The results indicated that these SDHIs cause multiple adverse effects in zebrafish embryos, including cardiovascular abnormalities, liver and kidney damage, and transcriptome changes. Chen et al. [7] studied isoflucypram cardiovascular toxicity in zebrafish, and Xiao et al. [8] investigated the chronic toxic effects of isoflufenacet on reproduction and intestinal energy metabolism in zebrafish. However, the toxicity, safety evaluation, and risk assessment of isoflucypram for humans have not been reported in the literature. Numerous scientific studies have documented the impact of drug binding with serum albumin on pharmacological and toxicological properties, making it a valuable tool for predicting in vivo toxicokinetics [9,10].

To investigate the impact of isoflucypram on the human body, HSA was chosen as the test substance. HSA is a highly abundant protein in blood plasma, with a concentration of approximately 35–50 g/L [11]. HSA is composed of a single chain and has a molecular weight of 66 kDa. It has a globular shape resembling a heart, with three similar domains labeled I, II, and III. Each domain consists of two helical subdomains that are connected by random coils [12,13]. Due to its unique structural and biochemical properties, HSA plays a crucial role in various physiological processes, including the transport of endogenous and exogenous ligands [14,15]. HSA has been shown to interact with a wide range of small molecules, including drugs, metabolites, and environmental toxins such as insecticides and fungicides [16,17]. Golianová et al. [16] Investigated the binding of conazole fungicides with serum albumins using spectroscopic methods complemented with molecular modeling. The interaction between the triazole fungicide tebuconazole and human serum albumin was investigated by Želonková et al. [17] using calorimetric and spectroscopic methods. These reports demonstrated that HSA can interact with pesticides, affecting their distribution, metabolism, and toxicity in the body [18].

In this study, molecular docking combined with molecular dynamics simulation was used to test the interaction between isoflucypram and HSA. Moreover, fluorescence spectroscopy was utilized to explore the underlying mechanisms, including the binding constant (K), thermodynamic parameters, and binding forces governing the interactions between isoflucypram and HSA under physiologically simulated conditions. In addition, we used UV-visible absorption (UV-vis), synchronous fluorescence, Fourier transform infrared (FT-IR), three-dimensional fluorescence (3D), and circular dichroism (CD) to discover the changes in the secondary structure of HSA in isoflucypram systems. Studying the interaction between HSA and isoflucypram is of great significance for the development of safer pesticides and the assessment of their exposure risks for humans.

## 2. Results and Discussion

### 2.1. Molecular Docking Studies

Figure 2 shows the best binding mode, which includes the interactions with amino acids surrounding the compound, as well as the bond distance. The HSA protein is represented as a slate cartoon model, and the ligand is shown as cyan sticks. The binding sites are depicted as magenta stick structures. Nonpolar hydrogen atoms are omitted. Hydrophobic interactions are represented as green dashed lines. Isoflucypram formed hydrophobic interactions with the Arg209 amino acid of HSA, with a bonding distance of 4.6 Å. Isoflucypram effectively bound to the active pocket of the protein with a binding energy of −8 kcal/mol, and its binding energy was less than −7 kcal/mol, indicating that the compound is capable of strong hydrophobic interactions with the protein binding pocket. Drug binding to HSA usually occurs at the Sudlow site I or II, which is a hydrophobic cavity that is capable of holding multiple drug molecules [19]. In light of the models of the complex of HSA with isoflucypram, the whole isoflucypram molecule may be localized in Sudlow’s site I of HSA. These results could serve as a structural foundation for further exploration of the binding mechanisms (e.g., fluorescence quenching and changes in thermodynamic parameters) between isoflucypram and HSA.

### 2.2. Molecular Dynamics Simulation Results

After a 100-ns molecular dynamics simulation of isoflucypram and HSA, a trajectory analysis was carried out. Firstly, the RMSDs of the trajectories of HSA and isoflucypram were extracted. As shown in Figure 3a, the protein and small molecule were in a relatively stable state after 85 ns. Regarding the observed variations in the RMSD value after 50 ns for the ligand in Figure 3a, we attribute this phenomenon to the inherent flexibility and dynamics of the ligand–protein complex. Throughout the simulation, the ligand may undergo slight conformational changes, resulting in fluctuations in the RMSD values. However, it is important to note that despite these variations, the overall stability and interaction between the ligand and protein remained relatively consistent. Therefore, we could undertake the following analysis for trajectories from 85 to 100 ns.

The RMSFs of track proteins and small molecules were extracted, as shown in Figure 3b,c. The green lines represent amino acid residues where small molecules form forces. The interaction mode of the stability interval (50–100 ns) of the kinetic trajectory was analyzed, as shown in Figure 3d. The amino acids that played important roles in small molecule binding included Phe206, Ala210, Lsy351, and Glu354, whose main roles are hydrophobic interactions, hydrogen bonding, and water bridges. The occupancy of interactions formed at 50–100 ns (number of frames forming interactions/total number of frames) was counted. The hydrophobic interaction of Lys351 led to occupancy of interactions at a frequency of 23%, indicating that the Lys351 amino acid played an important role in the binding process between isoflucypram and HSA.

### 2.3. Binding Mechanism

#### 2.3.1. Fluorescence Quenching Spectra

As displayed in Figure 4a, the HSA fluorescence emission spectra with (at 310 K) and without isoflucypram generated an obvious band absorption peak at 337 nm. The fluorescence intensity of HSA decreased with increasing isoflucypram concentrations, indicating that there was an interaction between the pesticide and HSA.

#### 2.3.2. Fluorescence Quenching Mechanism

Quenching is typically categorized into two types: dynamic quenching and static quenching, which differ in their sensitivity to temperature and viscosity [20]. To qualitatively explore the quenching mechanism of HSA by isoflucypram, the fluorescence data were analyzed in depth using the Stern–Volmer equation, and the following relationship was found to exist between F_0_/F and [Q] [21,22]:F_0_/F = 1 + K_sv_[Q] = 1 + K_q_τ_0_[Q](1)

The equation includes F_0_ and F, which represent the relative fluorescence intensities of HSA in the absence and presence of isoflucypram, respectively. K_sv_ represents the Stern–Volmer quenching constant. K_q_ indicates the quenching rate constant of the biomolecule. [Q] stands for the concentration of isoflucypram, and τ_0_ is the average lifetime of the molecule without the quencher (about 10^−8^ s).

Figure 4b shows the Stern–Volmer plots of the quenching of HSA fluorescence by isoflucypram at different temperatures. The calculated Stern–Volmer correlation and binding constants at different temperatures are listed in Table 1.

In the present system, the K_SV_ values increased with increasing temperatures, which indicated that HSA collided with isoflucypram and achieved energy transfer. The quenching of HSA by isoflucypram initiated the dynamic quenching process. One way to differentiate between static quenching and dynamic quenching is to consider the biomolecule’s limiting diffusion constant, which was determined to be 2.0 × 10^10^ L·mol^−1^·s^−1^. If the quenching constant (K_q_) is higher than 2.0 × 10^10^ L·mol^−1^·s^−1^, it indicates static quenching, whereas if K_q_ is lower than this value, it suggests dynamic quenching. Based on the data provided in Table 1, it can be inferred that the quenching mechanism between isoflucypram and HSA is partially static, as the K_q_ values were higher than 2.0 × 10^10^ L·mol^−1^·s^−1^ at 296 K, 303 K, and 310 K. Thus, the interaction mechanism between isoflucypram and HSA should be a dynamic and static mixing mechanism.

#### 2.3.3. Determination of Binding Constants and the Number of Binding Sites

The binding information between isoflucypram and HSA was obtained using the formula below, including the number of binding sites (n) and binding constants (K_a_) [23]:log[(F_0_ − F)/F] = logK_a_ + nlog[Q](2)

The equation relates the relative fluorescence intensities of HSA in the absence (F_0_) and presence (F) of isoflucypram, where K_a_ represents the binding constant of the interaction, n represents the number of binding sites, and [Q] represents the concentration of isoflucypram. By utilizing the formula, the log[(F_0_ − F)/F] versus log[Q] curve was generated through calculations (Figure 4c). The numbers of binding sites and binding constants at various temperatures were determined using a double-logarithm algorithm curve. The results of the calculated binding constants and binding sites are presented in Table 1.

The results suggested that the n values of isoflucypram were less than two, which may indicate that HSA had one binding site for isoflucypram. The binding constant decreased with increasing temperature. During the process of pesticide spraying, isoflucypram droplets are dispersed in the air or isoflucypram in material residues (in food or drinking water), entering the body through the mouth or nose, and are later absorbed into the bloodstream, where isoflucypram interacts with HSA during transport and metabolism.

#### 2.3.4. Thermodynamic Parameters and Binding Forces Analysis

The thermodynamic parameters, enthalpy change, entropy change, and free energy change are the main basis for determining the binding mode [24]. The Van’t Hoff equation was used to determine the main types of isoflucypram–HSA interactions and to calculate their thermodynamic constants [25]:lnK = −(ΔH^0^/RT) + ΔS^0^/R(3)
ΔG^0^ = ΔH^0^ − TΔS^0^(4)
where K is the binding constant at the respective temperature, R is the molar gas constant, and T is the experimental temperature. Utilizing the Van’t Hoff equation, the lnK vs. 1/T curve was obtained (Figure 4d). ΔG^0^, ΔH^0^, and ΔS^0^ were calculated using the slope and intercept. It can be seen from Table 2 that ΔG^0^ < 0, ΔH^0^ < 0, and ΔS^0^ < 0. The thermodynamic rules of the binding force between large molecules and small molecules showed that hydrogen bonding and van der Waals forces contributed greatly to the interaction when ΔH^0^ < 0 and ΔS^0^ < 0 [26]. For ΔG^0^ < 0, the interaction was thermodynamically favorable, i.e., the interaction tended to occur spontaneously.

#### 2.3.5. Energy Transfer and Binding Distance

The non-radiative energy transfer between isoflucypram and HSA was explained using the Förster energy transfer theory, and the binding distance was calculated [27]. The efficiency of energy transfer, the critical energy transfer distance, and the energy transfer distance between the donor and the acceptor had the following relationship:E = 1 − F/F_0_ = R_0_^6^/(R_0_^6^ + r^6^) (5)
where E represents the transfer efficiency between the donor and acceptor, r denotes the average distance separating the donor and acceptor, and R_0_ represents the critical distance at which the transfer efficiency reaches 50%. R_0_^6^ can be calculated using the following equation:R_0_^6^ = 8.8 × 10^−25^ K^2^N^−4^φJ (6)
where K^2^ denotes the spatial orientation factor of the dipole, N represents the refractive index of the medium, φ denotes the quantum yield of tryptophan in HSA, and J represents the overlap integral between the fluorescence emission spectrum of HSA and the absorption spectrum of carbamate pesticides. The calculation of J can be expressed using the following equation:J = ∑F(λ)ε(λ)λ^4^Δλ/∑F(λ)Δλ (7)
where F(λ) represents the corrected fluorescence signal of the donor within the wavelength range from λ to λ + Δλ, and ε(λ) denotes the extinction coefficient of the acceptor at wavelength λ. The quantum yield of tryptophan in HSA is 0.118 [28]. The average refractive index of the medium is 1.336, and the spatial orientation factor of the dipole, denoted as K^2^, is equal to 2/3 [29]. Figure 4e displays the fluorescence spectrum of HSA overlapped with the absorption spectrum of isoflucypram at 310 K, while Table 2 provides the binding distances. Thus, from the Eqs, we obtained the values of J, E, R_0_, and r (Table 2). Binding distances (r) of less than 8 indicated the occurrence of non-radiative energy transfer between isoflucypram and HSA. The results were consistent with the occurrence of a static quenching mechanism between isoflucypram and HSA.

### 2.4. Conformational Changes in HSA

#### 2.4.1. UV-Visible Absorption Spectra and Analysis

Using UV-vis absorption to investigate protein–drug interactions and ground state complex formation is a simple and readily accessible method [30]. According to the UV-vis spectra shown in Figure 5, HSA contained two distinctive absorption peaks located at approximately 214 nm and 278 nm. The modest absorption peak at approximately 278 nm was caused by the aromatic amino acids, and the significant absorption peak at approximately 214 nm was due to the absorption of the peptide skeleton structure of HSA (Trp, Tyr, and Phe) [31]. The UV-vis absorption signal of the peak at 214 nm exhibited a decrease in intensity with an increase in the concentration of isoflucypram. Additionally, the absorption signal of the peak at 278 nm did not change significantly. Notably, a red shift phenomenon was observed, whereby the maximum absorption wavelength shifted from 214 nm to approximately 216 nm. These results indicated that the polarity of the microenvironment near the peptide skeleton increased, and the microenvironment near the aromatic amino acids changed slightly. Distinguishing between dynamic and static quenching is possible using UV-vis spectra. Specifically, in dynamic quenching, the absorption spectrum of the fluorophore remains unchanged, whereas in static quenching, the absorption constantly changes [32]. Based on the UV-vis absorption spectrum of HSA with a range of isoflucypram concentrations, it can be inferred that the fluorescence quenching of HSA by isoflucypram was due to static quenching.

#### 2.4.2. Synchronous Fluorescence Spectra and Analysis

Synchronous fluorescence spectroscopy is a sensitive technique for probing changes in ligand binding pocket conformation and polarity [33]. Miller [34] proposed a long-held theory that, when Δλ is stabilized at 15 nm and 60 nm, the synchronous fluorescence of HSA exhibits the characteristic information of Tyr and Trp residues, respectively. Because the maximum emission wavelength of amino acid residues might reflect the hydrophobicity of their surrounding environment, the influence of pesticides on HSA conformation could be established by the change in emission wavelength [25]. Based on Figure 6, it can be observed that the maximum fluorescence emission peak of the protein decreased as the isoflucypram concentration increased under the spectrum of Δλ = 15 nm and Δλ = 60 nm. This indicated that the fluorescence of the Tyr and Trp residues in HSA was quenched, and there was a slight change in the microenvironment surrounding these residues.

#### 2.4.3. Three-Dimensional Fluorescence Spectra and Analysis

Protein conformational changes can be elucidated using three-dimensional (3D) fluorescence spectrum analysis [35]. Figure 7 displays the three-dimensional fluorescence spectra of conformational changes in HSA both before and after the addition of isoflucypram (a and b, respectively). The spectrum properties of the tryptophan and tyrosine residues were mainly revealed in peak 1, whereas the fluorescence spectral behavior of the polypeptide backbone structures was mainly demonstrated in peak 2. The fluorescence signal of peak 1 clearly decreased with the addition of isoflucypram, showing an increase in the polarity of the environment around the tryptophan and tyrosine residues. The fluorescence signal of peak 2 decreased, suggesting that the polypeptide structure of HSA had been altered.

#### 2.4.4. FT-IR Spectra and Analysis

Infrared spectroscopy can reveal the amide bands of proteins and uncover the vibrations of peptides. For the characterization of protein structural changes, especially the determination of secondary structures, FT-IR spectroscopy is widely used [27]. Since the intensity of the amide I band demonstrates sensitive dependency on the protein backbone, several studies have concentrated on this band [36,37]. IR band frequencies resulting from amide I band (1600–1700 cm^−1^) vibrations provide a large amount of information about the secondary protein structure. The amide I peak has been further divided into four structural units: α-helix (1650–1658 cm^−1^), β-sheet (1610–1640 cm^−1^), β-turn (1660–1700 cm^−1^), and random coil (1640–1650 cm^−1^) [38].

The FT-IR fitting curves for HSA in the presence and absence of ligands are displayed in Figure 8. With the addition of isoflucypram, the peak at 1610.96 cm^−1^ shifted to 1611.74 cm^−1^, the peak at 1623.22 cm^−1^ shifted to 1625.48 cm^−1^, the peak at 1634.95 cm^−1^ shifted to 1638.07 cm^−1^, the peak at 1645.22 cm^−1^ shifted to 1648.13 cm^−1^, the peak at 1655.67 cm^−1^ changed to 1657.71, the peak at 1666.79 cm^−1^ changed to 1668.95 cm^−1^, and the peak at 1679.42 cm^−1^ increased to 1684.49 cm^−1^. The positions of these peaks were red-shifted, indicating that the secondary structure of HSA was altered by the addition of ISO. These adjustments in peak position showed that ligand binding to HSA had some effect on the protein’s structural dynamics. Due to interactions with the addition of isoflucypram (Table 3), the content of the α-helix clearly dropped from 22.04% to 15.22%. In contrast, the content of the β-turn, β-sheet, and the random coil increased from 33.63% to 39.35%, 27.49% to 28.24%, and 16.84% to 17.19%, respectively. These results indicated that with the addition of isoflucypram, the secondary structure of HSA was altered from an α-helix to a β-turn, β-sheet, and random coil, and the structure of HSA loosened. The above findings further clarified that isoflucypram and HSA can be combined.

#### 2.4.5. CD Spectrum and Analysis

Circular dichroism (CD) spectroscopy is a valuable method for analyzing changes in the secondary structure of proteins [39]. As shown in Figure 9, the CD spectrum of HSA exhibited a negative Cotton effect at 208 nm and 222 nm, indicating the presence of an α-helical structure, which is a typical secondary structure of proteins. However, the negative Cotton effect of HSA at these wavelengths was diminished with the addition of isoflucypram. This decrease suggested a change in the protein’s secondary structure induced by the drug. Although the negative Cotton effect of HSA was reduced by isoflucypram, the shape of the CD spectrum did not change, indicating that the protein’s secondary structure was still dominated by the α-helix structure. The mean residue ellipticity (MRE) is a commonly used parameter for quantifying the magnitude of protein CD spectra. It can provide valuable information on the degree of structural change induced by a drug or other perturbations. The mean residue ellipticity (MRE) in the following formula is commonly used to represent the extent of protein circular dichroism in the study of circular dichroism:MRE = ObservedCD(mdeg)/(10 Cpnl)(8)

In this equation, C_p_ is the molar concentration of HAS (2.0 × 10^−6^ M), n is the number of amino acid residues (585 for HSA), and l is the path length (1 mm). The α-helix contents of the free and bound HSA were calculated from MRE values at 208 nm using the following equation [40]:α − helix(%) = [(−MRE208 − 4000) × 100]/(33,000 − 4000) (9)

In the equation, MRE_208_ is the actual MRE value measured at 208 nm, 4000 refers to the MRE value of the random coil conformation at 208 nm, and 33,000 is the MRE value of a pure α-helix at 208 nm.

Figure 9 shows the CD spectra of HSA at 200–260 nm, with and without isoflucypram. By using the above formula, we found that the α-helix content of HSA was calculated to be reduced from 55.85% to 52.12% and 42.66% at isoflucypram/HSA molar ratios of 5:1 and 10:1, respectively. The decreasing α-helix content suggested that the binding of isoflucypram to HSA induced a modicum of unfolding in the constituent polypeptides, thereby changing the secondary HSA structure. HSA is the most significant protein for storage and transport in the human blood circulatory system [41]. Alterations in the secondary structure of HSA may affect the transport and metabolism of other compounds in the blood.

## 3. Materials and Methods

### 3.1. Reagents and Chemicals

Isoflucypram was obtained from the Dr. Ehenstorfer GmbH company (Augsbur, Germany). HSA (purity 96%, with a molar mass of 66.5 kDa) was purchased from Beijing Solarbio Science & Technology Co., Ltd. (Beijing, China) and dissolved in 0.02 M phosphate-buffered saline (PBS, pH 7.4), which was bought from Leagene Biotechnology Corporation (Beijing, China) and stored at 4 °C in the dark. The maximum amount of ethanol added to the HSA solution was 3% to ensure that it would not have an impact on the HSA. The water used in all of the tests was double distilled, and all the other reagents were of analytical grade.

### 3.2. Molecular Docking

Molecular docking is one of the most important tools for studying drug–protein interactions [42,43]. Molecular docking technology is based on the “Key–Lock principle” [44], in which rigid docking, semi-flexible docking, and flexible docking are the main molecular docking methods. The X-ray crystal structure of HSA was downloaded from the RCSB Protein data bank (http://www.rcsb.org/ (accessed on 6 July 2023)) with PDB ID: 1AO6 at a 2.50 Å resolution. The protonation state of isoflucypram was set at pH = 7.4, and the compounds were expanded to 3D structures using Open Babel [45]. The HSA had the water molecules removed, polar hydrogen was added, Gasteiger charges were added, and the atom type was assigned as the AD4 type by AutoDock Tools (ADT3). Isoflucypram was identified in ligand pdb files after following the same steps as those for HSA. The grid spacing parameter (1 Å) and a grid box of 50, 48, and 52 Å were used to investigate the most likely binding position. AutoDock Vina (1.2.0) was used to run the blind docking calculations [46,47]. The number of modes was set to nine to find the most likely binding position. After the analysis, the most reasonable site for the binding of isoflucypram and HSA was selected. We selected Pymol to visualize the complex [48].

### 3.3. Molecular Dynamics Simulation

The molecular dynamics simulations were carried out using Desmond/Maestro noncommercial version 2022.1 as the molecular dynamic software [49]. TIP3P water molecules were added to the systems, which were then neutralized by a 0.15 M NaCl solution. After the minimization and relaxation of the system, the production simulation was performed for 100 ns in an isothermal–isobaric ensemble at 310 K and 1 bar. The trajectory coordinates were recorded every 100 ps. The molecular dynamics analysis was performed using the Simulation Interaction Diagram from Desmond.

### 3.4. Fluorescence Spectroscopy

All the fluorescence spectra were obtained using a FL6500 fluorescence spectrophotometer (PerkinElmer, America) equipped with 1.0 cm quartz cells. Fluorescence titrations were conducted by fixing the HSA concentration at 5 × 10^−7^ M while varying the isoflucypram concentration from 0 to 105 × 10^−7^ M. An excitation wavelength of 280 nm was employed, and the emission spectra were recorded from 300 nm to 500 nm. The excitation and emission slit widths were 20 nm. The fluorescence signals of a series of assay samples were recorded after preheating for 2 min at 296 K, 303 K, and 310 K. The following equation was used to account for the inner filter effect in the fluorescence signals [50,51]:F_cor_ = F_obs_exp[A_ex_ + A_em_/2] (10)

In this formula, Fcor and Fobs are the corrected and observed fluorescence signals, respectively, and A_ex_ and A_em_ are the system absorbances at excitation and emission wavelengths, respectively.

### 3.5. UV-Vis Absorption Spectroscopy

UV-vis absorption spectra were acquired using a UV-vis spectrophotometer (UV1800, Shimadzu, Japan) with 10 mm quartz cells. The UV-vis spectra of HSA with the addition of different isoflucypram concentrations were recorded in the range of 200–320 nm at 310 K. The experiment was performed by keeping the concentration of HSA constant at 5.0 × 10^−6^ M while varying the concentration of isoflucypram from 0 to 175 × 10^−6^ M at 310 K. The wavelength interval was set to 0.5 nm, and the slit width was set to 2 nm.

### 3.6. Synchronous Fluorescence Spectroscopy

Synchronous fluorescence spectra of HSA in the absence and presence of different isoflucypram concentrations were recorded by scanning the excitation and emission spectra simultaneously, in the range of 200 to 300 nm at 310 K. The wavelength intervals (Δλ) between excitation and emission were set to 15 nm and 60 nm, respectively. The concentration of HSA was kept at 2 × 10^−6^ M with 0 to 18 × 10^−6^ M pesticide when Δλ = 15 nm, and was kept at 2 × 10^−6^ M with 0 to 48 × 10^−6^ M pesticide when Δλ = 60 nm.

### 3.7. Three-Dimensional Fluorescence Spectroscopy

The fluorescence spectra of a mixture of 1 × 10^−4^ M HSA and 1 × 10^−2^ M isoflucypram were measured at 310 K using a three-dimensional fluorescence spectrometer. The emission wavelength was recorded from 250 nm to 500 nm. The slit widths for excitation and emission were both 20 nm, and the initial excitation wavelength was set at 210 nm with a 1 nm step.

### 3.8. Fourier Transform Infrared Spectroscopy

FT-IR spectra were recorded on a Nicolet IS5 FT-IR spectrometer (Thermo Fisher Scientific, Waltham, MA, USA). Prior to the FT-IR spectroscopy measurements, the solutions were preheated at 310 K for 3 min. In a buffer of 0.02 M PBS, the FT-IR spectra of HSA (2.0 × 10^−4^ M) in the absence and presence of isoflucypram (2.0 × 10^−4^ M) were collected, and background spectra were collected before each measurement. At a resolution of 4 cm^−1^ and an average of 64 scans, all the spectra were collected between 1800 and 700 cm^−1^.

### 3.9. CD Spectroscopy

CD spectra were collected using a CD spectrometer (Chirascan, Applied Photophysics Ltd., UK). The spectra were recorded in a 1 mm path-length cell under a nitrogen atmosphere. The CD spectra of HSA were recorded at a wavelength between 200 nm and 260 nm in the absence and presence of isoflucypram (pH = 7.4) at 310 K. The HSA concentration was kept constant at 2 × 10^−6^ M, and the ratios of pesticide to HSA were 0:1, 5:1, and 10:1.

## 4. Conclusions

The molecular docking and molecular dynamics simulation results showed that the HSA binding of isoflucypram was due to binding forces that included hydrophobic interactions, hydrogen bonding, and water bridges. The multispectral research focused on two aspects: a discussion of the interaction mechanisms and a conformational alteration study. The mechanism of the interaction between isoflucypram and HSA was mixed (static and dynamic) quenching. The distance (r) between the binding position of isoflucypram and HSA was 20.17 Å. The interaction between isoflucypram and HSA tended to occur spontaneously in thermodynamics. UV-visible absorption, synchronous fluorescence, and 3D fluorescence spectra supported the idea that the Tyr and Trp microenvironment near the aromatic amino acids of HSA and the polypeptide structure changed slightly. CD spectra and FT-IR spectra showed that the α-helix content of HSA was reduced, and the β-turn, β-sheet, and random coil contents were increased after the addition of isoflucypram. These results further elucidated that isoflucypram can bind with HSA. This study suggests that human exposure to isoflucypram may affect metabolism by influencing the structure of HSA. Finally, these research results support our assessment of the toxicity risks, showing that isoflucypram exposure poses a risk to human health, which can inform future research.

## Figures and Tables

**Figure 1 ijms-24-12521-f001:**
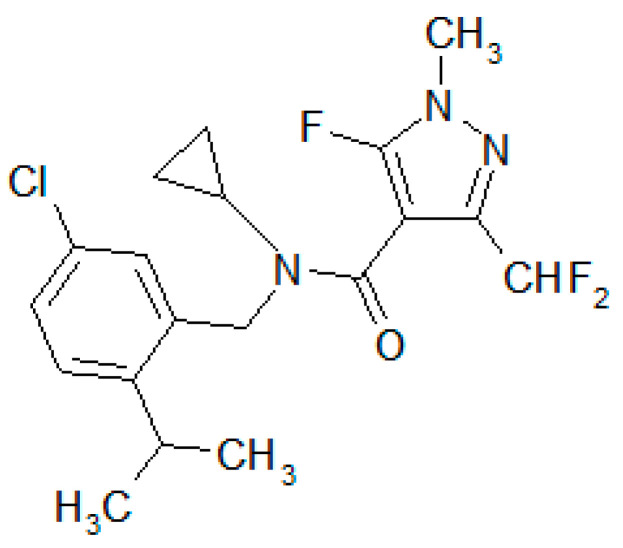
Chemical structure of isoflucypram.

**Figure 2 ijms-24-12521-f002:**
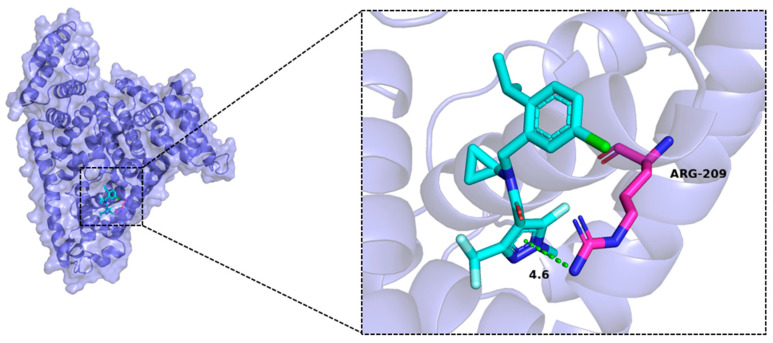
Schematic diagram of the optimal isoflucypram–HSA conformation generated by AutoDock Vina 1.1.2.

**Figure 3 ijms-24-12521-f003:**
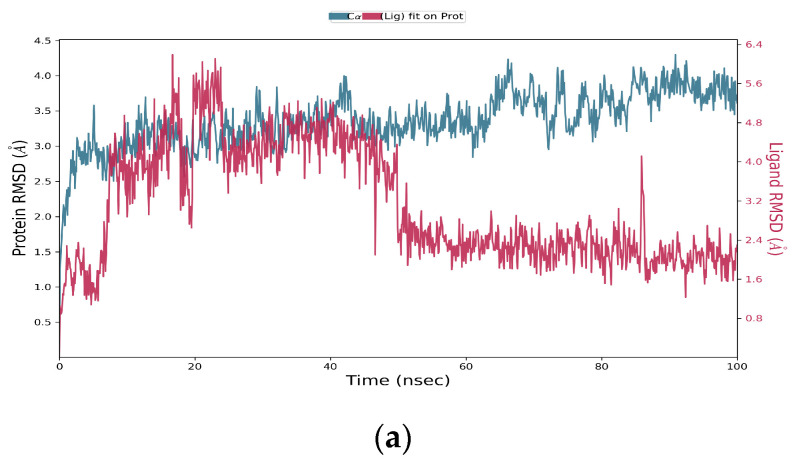
Molecular dynamics simulation results for isoflucypram and HSA. (**a**) RMSD; (**b**) HSA RMSF; (**c**) isoflucypram RMSF; (**d**) the connection between the ligand and protein.

**Figure 4 ijms-24-12521-f004:**
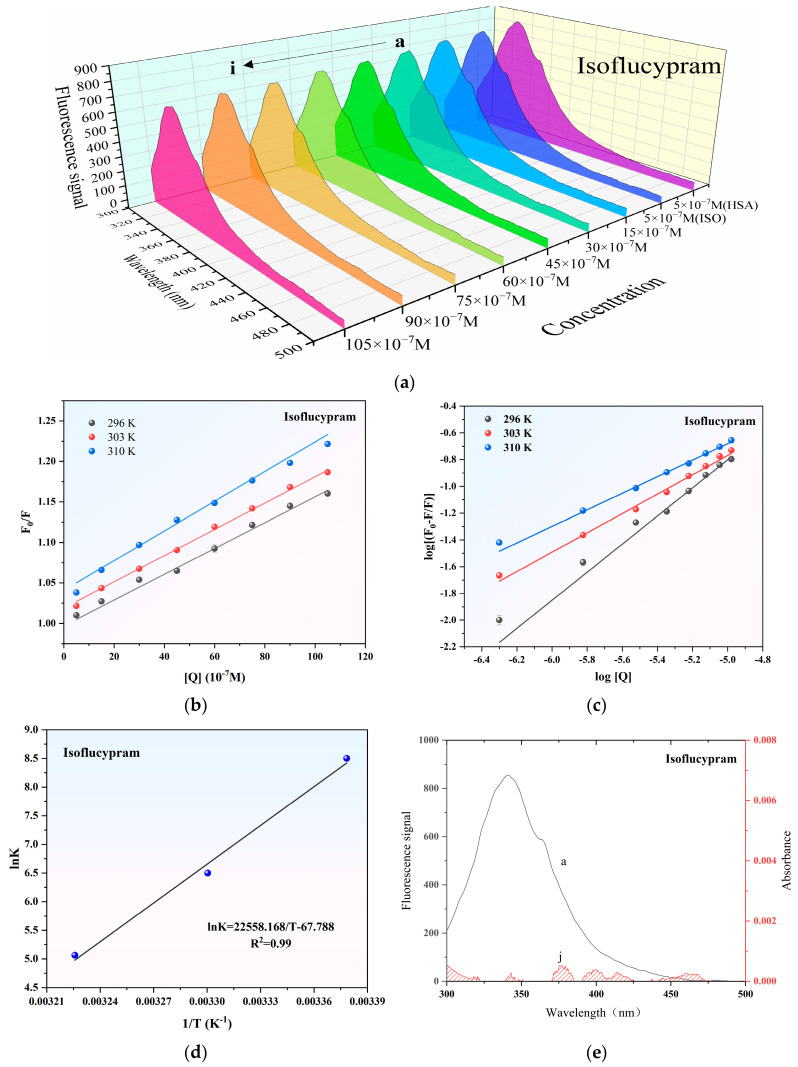
Fluorescence spectroscopy of the interaction between isoflucypram and HSA. (**a**) Fluorescence emission spectra of the interaction between isoflucypram and HSA. (**b**) Stern–Volmer plots of the interaction. (**c**) Plots of log[(F_0_ − F)/F] versus log[Q] of the interaction. (**d**) Plot of lnK versus 1/T based on the Van’t Hoff equation. (**e**) HSA’s fluorescence spectrum overlaps with isoflucypram’s UV-vis absorption spectrum. (a–i) 5 × 10^−7^ M HSA with the presence of 0–105 × 10^−7^ M isoflucypram; (j) 5 × 10^−7^ M isoflucypram; [Q]—isoflucypram concentration; F_0_/F—HSA fluorescence signal in the presence or absence of isoflucypram. K—binding constant (K_a_); T—absolute temperature, pH = 7.4.

**Figure 5 ijms-24-12521-f005:**
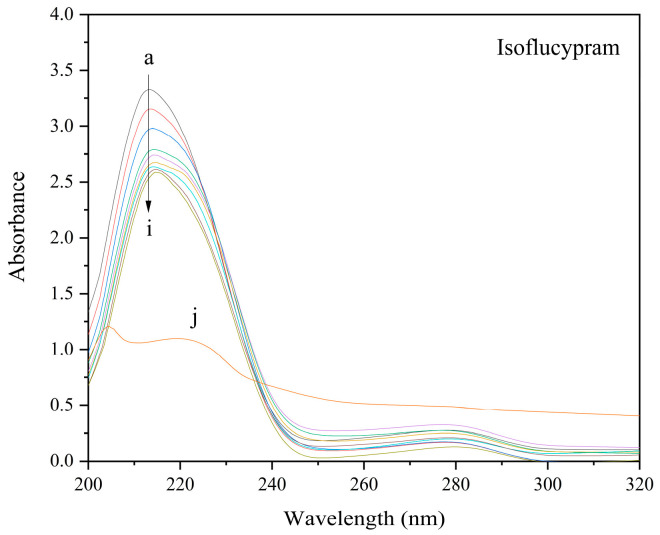
Ultraviolet-visible spectra of HSA in the presence of various concentrations of isoflucypram. (a) 5 × 10^−6^ M HSA only; (b–i) 5 × 10^−6^ M HSA in the presence of 5, 25, 50, 75, 100, 125, 150, and 175 × 10^−6^ M isoflucypram; (j) 5 × 10^−6^ M isoflucypram only, pH = 7.4, T = 310 K.

**Figure 6 ijms-24-12521-f006:**
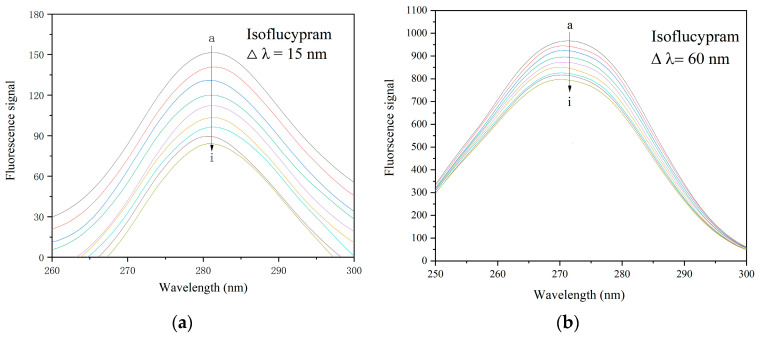
Synchronous fluorescence spectrum of HSA in the absence and presence of a range of concentrations of isoflucypram. (a–i) HSA with various concentrations of isoflucypram. The HSA concentration is 2 × 10^−6^, pH = 7.4, T = 310 K. (**a**) When Δλ is 15 nm, the ratio of isoflucypram to HSA is 0:1, 2:1, 3:1, 4:1, 5:1, 6:1, 7:1, 8:1, and 9:1. (**b**) When Δλ is 60 nm, the ratio of isoflucypram to HSA is 0:1, 3:1, 6:1, 9:1, 12:1, 15:1, 18:1, 21:1, and 24:1.

**Figure 7 ijms-24-12521-f007:**
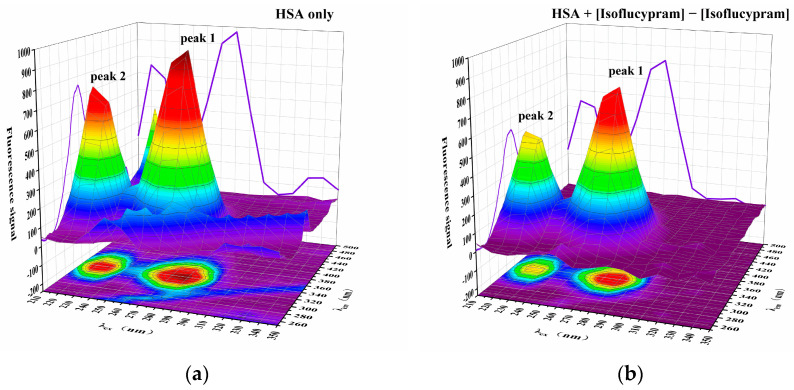
Three-dimensional fluorescence spectra of HSA in the absence and presence of isoflucypram. (**a**) 1 × 10^−4^ M has; (**b**) 1 × 10^−4^ M HSA with 1 × 10^−2^ M isoflucypram, pH = 7.4, T = 310 K.

**Figure 8 ijms-24-12521-f008:**
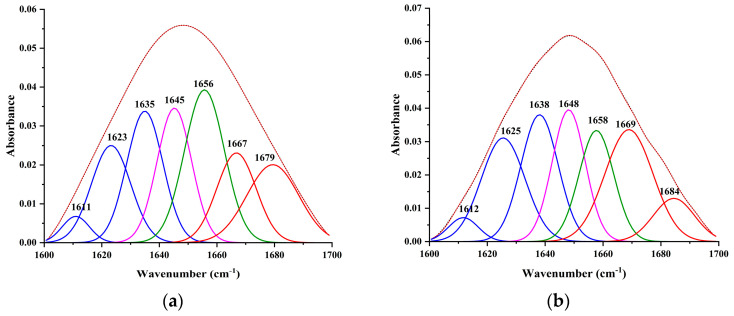
Curve-fitting of amide I region (1700–1600 cm^−1^) and determination of the secondary structure of free HAS. (**a**) 4 × 10^−4^ M HSA only; (**b**) 4 × 10^−4^ M HSA. The molar ratio of isoflucypram to HSA was 1:1, pH = 7.4, T = 310 K.

**Figure 9 ijms-24-12521-f009:**
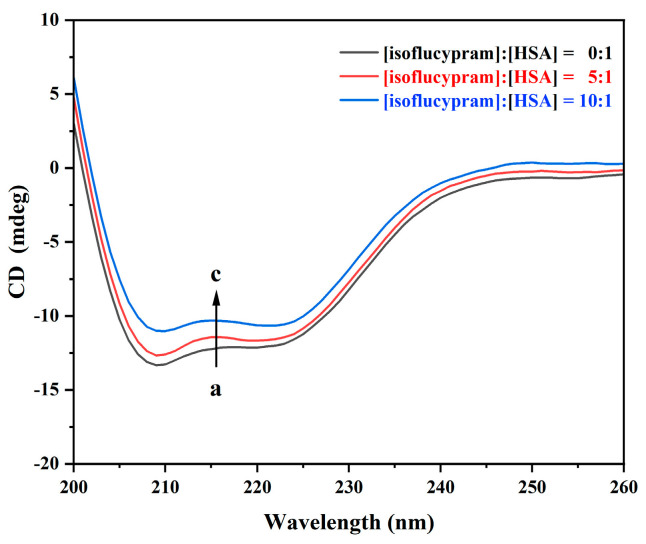
Circular dichroism spectra of HSA with various concentrations of isoflucypram. (**a**) 2 × 10^−6^ M HSA only, (**b**–**c**) 2 × 10^−6^ M HSA. The molar ratios for isoflucypram to HSA were 5:1 and 10:1, pH = 7.4, T = 310 K.

**Table 1 ijms-24-12521-t001:** Stern–Volmer correlation constants and binding constants associated with the HSA–isoflucypram systems, as determined using fluorescence spectroscopy.

T (K)	K_sv_ (10^4^ M^−1^)	R^2^	K_q_ (10^12^ M^−1^·s^−1^)	n	K_a_ (10^3^ M^−1^)
296 K	1.593 ± 0.073	0.994	1.593 ± 0.073	0.901 ± 0.011	4.923 ± 0.619
303 K	1.619 ± 0.027	0.999	1.619 ± 0.027	0.803 ± 0.006	0.664 ± 0.037
310 K	1.832 ± 0.076	0.995	1.832 ± 0.076	0.577 ± 0.006	0.158 ± 0.011

**Table 2 ijms-24-12521-t002:** Thermodynamic parameters, binding distance, and non-radiative energy transfer of the HSA–isoflucypram system, as determined using fluorescence spectroscopy.

T (K)	ΔH^0^ (KJ·mol^−1^)	ΔS^0^ (J·mol^−1^·K^−1^)	ΔG^0^ (KJ·mol^−1^)	J (cm^3^·L·mol^−1^)	R_0_ (nm)	E	r (nm)
310 K	−187.549	−563.59	−12.836	1.183 × 10^−17^	1.170	3.67%	2.017

**Table 3 ijms-24-12521-t003:** Average percentages of the secondary structures of HSA after the addition of isoflucypram.

Secondary Structures (%)	Only HSA	Isoflucypram + HSA
α-helix	22.04%	15.22%
β-sheet	33.63%	39.35%
β-turn	27.49%	28.24%
random coil	16.84%	17.19%

## Data Availability

The data are contained within the article.

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
