# Peer review of "Probing the Interaction between Isoflucypram Fungicides and Human Serum Albumin: Multiple Spectroscopic and Molecular Modeling Investigations"

_ijms, 2023, doi:10.3390/ijms241512521_

Round 1
Reviewer 1 Report
In introduction, there are only few references about the papers dealing with the interactions of pesticides with HSA. For example, fungicides tebuconazole, prothioconazole, epoxiconazole interact with HSA - similar results like yours are in:
GOLIANOVÁ, Katarína - HAVADEJ, Samuel - VEREBOVÁ, Valéria - ULIÄŒNÝ, Jozef - HOLEÄŒKOVÁ, Beáta - STANIÄŒOVÁ, Jana. Interaction of conazole pesticides Epoxiconazole and Prothioconazole with human and bovine serum albumin studied using spectroscopic methods and molecular modeling. In: International Journal of Molecular Sciences. 2021. ISSN 1422-0067, Vol. 22, no. 4 (2021), art. no. 1925, p. [1-19]
and
ŽELONKOVÁ, Katarína - HAVADEJ, Samuel - VEREBOVÁ, Valéria - HOLEÄŒKOVÁ, Beáta - ULIÄŒNÝ, Jozef - STANIÄŒOVÁ, Jana. Fungicide tebuconazole influences the structure of human serum albumin molecule. In: Molecules. 2019. ISSN 1420-3049, Vol. 24, no. 17 (2019), art. no. 3190, p. [1-15]
and insecticides are comprehensively summarized in:
VEREBOVÁ, Valéria - STANIÄŒOVÁ, Jana. The effect of neonicotinoid insecticides on the structure and stability of bio-macromolecules. In: RANZ, Ramón Eduardo Rebolledo et al. Insecticides : Impact and Benefits of Its Use for Humanity. 1. vyd. London : IntechOpen, 2022. ISBN 978-1-83969-026-6, online, s. 245-270 [2,19AH]
Major remarks:
1. The title does not contain the fact, that isoflucypram is pesticide (fungicide)
2. Where exactly is isoflucypram localizated in HSA? In Sudlow I or II? In which structural domain of HSA? By aminoacids, it looks like Sudlow II.
3. Equation (5) in literature: E = 1 - F/F0 , does not correlate with yours eq. (5)
4. Original FTIR spectra are missing, only deconvolutions are in the manuscript. I recommend to include these spectra into manuscript.
5. Section 3. Discussion should be 4. Conclusion.
6. In row 379, there is a binding distance 201.7 A°, which is bad transformation between nm and angstrom. It should be 20.17 A°.
Minor remarks:
row 3: in title: The words should be written with have capital letter; The title is not finished because spectroscopic what? methods?
row 10 and others: Isoflucypram - isoflucypram
row 28: However, A = a; row 36 and others: et al = et al. ; rows 49, 50: HAS = HSA; row 62: This = this; row 83: kcal = kJ; rows 86, 88: bonding distances 3.1 and 3.3, 3.6 = unit is missing; row 88: In addition to; row 95: Figure 2 contains unreading text, Rearrange the figure; row 110: equations should be in Italic, even in the text, symbols should be in Italic; row 111: bad description of variables in equation (1); Figure 3d: LnK = lnK; row 130: One way = one way; rows 142, 145: F0 = F0 ; row 161: LnK = lnK; row 169: Table: 10_17 = 10-17 and the values from the second row should be shifted to the third row (for 310 K); row 177: Equation 5: R06 = R0 6 ; row 189: at 310 K, in Table 2, the values are for 303 K - see row 169; row 190: Eqs = what equations?; row 214: Figure is not finished; row 272: The = the; row 287: Cp = Cp ; row 304: The = the; row 326: Bad title: Molecular docking = Fluorescence spectroscopy; row 330: 280nm = 280 nm; rows 335,336: indeces!; row 352: HAS = HSA; rows 371,372: Sentences do not make meaning; row 381: , Synchr. = , synchr.; row 387: , The = the
Reviewer 2 Report
The paper entitled “Analysis of binding properties and interaction of Isoflucypram with HSA via multiple spectroscopic” by Xiangshuai Li et al. provides the structure-related study of the next-generation succinate dehydrogenase inhibitor fungicide with the most abundant protein in plasma (Human Serum Albumin). On the whole, the conjugation of in-silico methodology with the experimental procedures (spectroscopic methods) might be appealing to the scientific community and the obtained findings are within the scope of the IJMS journal. On the other hand, the quality of the paper is definitely below the basic requirements that should be fulfilled by scientific papers. In my humble opinion, many major issues prevent the reviewed text from suggesting for publication in IJMS.
Let me introduce just some of them – there are too many to list them all in details.
1. First of all, I strongly encourage the Authors to rearrange the text of the manuscript, because it is vaguely written that makes it difficult to understand it properly. Definitely, the quality of language must be improved, because there are so many errors (grammar, stylistic, word repetitions, etc.). Extensive editing of English language is required, because it is completely incomprehensible, in particular the introductory part (abstract and introduction). For instance, lines 62-64 ‘At first, This study used molecular docking approach the binding sites of Isoflu-62 cypram on HSA were predicted and show that the distances between Isoflucypram and 63 amino acid residues.’ or line 49 ‘HAS’ and many, many others.
2. I do not understand the difference between ‘binding properties and interaction’ as stated in the title of the paper.
3. What means ‘Computational molecular docking analysis can elucidate how active site binding occurs and can also provide a target orientation to experimental results [16].’ In what sense ‘to experimental results’?
4. The whole part describing the details of the molecular docking ligand/enzyme preparation is missing. Have Authors followed just the typical procedure implemented in AutoDock Vina?
5. The Authors stated that ‘the results of these docking simulations suggest that Isoflucypram forms stable complexes with HSA and that the forces at different positions of Isoflucypram on HSA play an important role in stabilizing the Isoflucypram-HSA complex.’ The above statement should be confirmed by the MD simulations of the Isoflucypram-HSA complex. Has it been done?
6. What was the reason that the apo (liganded-free) HSA 3D crystalline conformation with a pretty poor resolution of 3.23 A was downloaded from PDB database.
7. Which aminoacids were selected to perform the semi-flexible docking simulations?
8. How was specified the binding site of the HSA? I have not noticed the SITE specification inside the pdb file.
9. The references must be unified according to the journal requirements.
I strongly encourage the Authors to rearrange the text of the manuscript, because it is vaguely written that makes it difficult to understand it properly. Definitely, the quality of language must be improved, because there are so many errors (grammar, stylistic, word repetitions, etc.).
Round 2
Reviewer 2 Report
The revised version of the paper entitled “Probing the Interaction of Isoflucypram Fungicides with HSA: Multiple Spectroscopic and Molecular Modeling Investigations” by Xiangshuai Li et al. provides the structure-related study of the next-generation succinate dehydrogenase inhibitor fungicide with the most abundant protein in plasma (Human Serum Albumin). Basically, the conjugation of in-silico methodology with the experimental procedures (spectroscopic methods) might be appealing to the scientific community and the obtained findings are within the scope of the IJMS journal. I appreciate an effort of the Authors to follow my previous suggestions and recommendations; however there are still some issues that prevent the revised text from suggesting for publication in IJMS.
Despite the Authors’ reassurance I still strongly encourage them to rearrange the text of the manuscript, because there are so many errors (grammar, stylistic, word repetitions, etc.). Extensive editing of English language is still required, probably by a native speaker. For instance, many times the same words are repeated in lines:|
41,42 ‘studied’
77,78 ‘showed’
120, 122 ‘represents’
124,125 ‘shows’
259 ‘reveal’
292, 293 ‘decrease’
172 ‘can be obtained’
Lines 30,322,339 ‘isoflucypram’
Line 100 ‘acidsthatplay’
Line 103 ‘offrames’
Line 127 ‘Binding’
Line 263 missing space
In line 44 the sentence seems to be not finished.
Line 59 ‘Insecticides’
Lines 69-70, upper letters are used unnecessarily
Line 249 A or a, B or b?
Line 271 Figure 8 or Fig. 8. It should be unified in the manuscript
Line 338 ‘Gasteriger’
Line 342 ‘ofmodes’
Line 309 ‘were calculated reduced’
and others.
2. Authors stated in lines 79-81 that ‘isoflucypram can effectively bind to the active pocket of the protein with a binding energy of -8 kcal/mol, and its binding energy is less than -7 kcal/mol’. Could you explain the difference between values of binding energy?
3. Authors stated in lines 94-95 that ‘protein and small molecule were in a mutually stable state after 50ns’, but in Figure 3a a relatively huge variations of RMSD value for ligand is observed. Could you provide any explanation of the observed phenomenon?
4. Conclusion section should be clarified, for instance, what means ‘the idea that the Tyr and Trp microenvironment surrounding the microenvironment near the aromatic amino acids of HSA’?
5. The references must be unified according to the journal requirements, because Authors use (or not) dots in the journal abbreviations (Ref. [4] or Ref. [3]), do not include number (Ref. [3], [11]), etc.
Despite the Authors’ reassurance I still strongly encourage them to rearrange the text of the manuscript, because there are so many errors (grammar, stylistic, word repetitions, etc.). Extensive editing of English language is still required, probably by a native speaker.
Round 3
Reviewer 2 Report
In general, the Authors have followed my suggestions. I still see some minor mistakes, but I hope it will be corrected by IJMS editors.
I still see some minor mistakes, but I hope it will be corrected by IJMS editors.